# Utilization of TiN and the Texture of Bionic Pangolin Scales to Improve the Wear Resistance of Cast Steel 20Mn Metal

**DOI:** 10.3390/biomimetics10010042

**Published:** 2025-01-10

**Authors:** Wenwen Zhang, Mingyuan Zhang, Xingliang Dong, Yuanzhe Huang, Shukun Cao

**Affiliations:** School of Mechanical Engineering, University of Jinan, Jinan 250022, China; zwenwen202207@163.com (W.Z.); me_zhangmy@ujn.edu.cn (M.Z.); 15854103370@163.com (X.D.); hyuanzhe2022@163.com (Y.H.)

**Keywords:** bionic pangolin, micro-textures, TiN coating, sliding wear

## Abstract

This research centers around cast steel 20Mn, which is the material utilized for the ear-picking roller of a corn harvester. The study delves into methods of enhancing its hydrophobicity and wear resistance. Fiber laser-processing technology was employed to fabricate pangolin bionic micro-textures on the material surface, and PVD technology was utilized to deposit a TiN coating. The wear resistance of the modified surface was investigated via the reciprocating dry sliding wear method, while its hydrophobicity was concurrently examined. The results demonstrate that the laser texturing technology and TiN coating managed to reduce the friction coefficient of the sample surface by 20% and 30.9%, respectively. This can be chiefly attributed to the significant effects of the modified surface in augmenting hardness, diminishing the contact area of the friction surface, lowering shear stress, and entrapping wear debris. When the pangolin texture and TiN coating work in concert, the abrasive and fatigue wear between the two surfaces is conspicuously mitigated, and the friction coefficient is reduced by 38.09%. Moreover, the experiment also reveals that a superhydrophobic surface can be achieved by fabricating the pangolin micro-textures.

## 1. Introduction

Cast steel 20Mn serves as a crucial material for the ear-picking rollers of corn harvesters, playing a vital role in detaching the stems from the ears of crops during the harvest process. This particular component is subject to high-speed rotation, substantial load, and an extremely harsh working environment throughout the crop harvesting period. Consequently, it is essential for it to possess high wear resistance. Thus, research focused on enhancing the wear resistance of the material employed in the ear-picking rollers of corn harvesters has emerged as a highly significant and actively pursued area of study.

Currently, there are many methods to improve material wear resistance, such as surface micro-texturing [1], surface coatings [2], heat treatment [3], etc. Surface micro-texturing involves processing a series of shapes on the material surface to alter its surface properties. Methods for surface micro-texturing include abrasive jet machining [4], laser processing [5,6], electrical discharge machining [7], plasma etching [8], mechanical processing, embossing and die forging [9], among others. Among these technologies, the laser surface texture process is considered a promising method, which has the characteristics of high operation accuracy, shallow heat-affected zone, and low cost [10,11]. Del et al. [12] used a nanosecond Yb-fiber laser system to process parallel grooved micro-texturing on high reinforcement concentration AlSiC-9 Metal Matrix Composite and conducted friction and wear tests. The results showed a 60% reduction in friction coefficient and a 70% reduction in wear, and SEM/EDX analysis revealed adhesion, abrasion, and plastic deformation as the main wear mechanisms.

Many studies have explained the effectiveness of Laser microtexture in improving material friction and wear performance by introducing geometric micro-patterns (such as dimples, grooves, and grids) on the surface. Further research has shown that besides some standard patterns, biomimetic textures offer better research potential. For example, Li et al. [13] used femtosecond lasers to process three types of biomimetic textures on TC4 samples: snake scales, fish scales, and shell structures. They studied the wear resistance of the textured surface through rotational dry friction and found that the textured surface effectively reduced and captured the generation of wear particles, enhancing the stability of TC4 during the wear process and reducing surface wear. Compared to a smooth surface, the wear on the textured surface was reduced by 73.68%.

Apart from surface texturing, surface coating is also a common method for improving material wear resistance. Methods for processing surface coatings include spraying, chemical vapor deposition [14], physical vapor deposition [15], and electrochemical deposition. The materials used for coatings are typically aluminum oxide (Al_2_O_3_), tungsten carbide (WC) [16], titanium nitride (TiN) [17], among others. TiN coatings have relatively high hardness and good wear resistance and chemical stability, so many researchers are enthusiastic about using various methods to process TiN coatings onto matrix materials to improve the specific properties of TiN coatings [18,19,20].

Currently, as research progresses, engineers are gradually combining coatings with textures to study the synergistic effects of coatings and textures on the tribological properties of materials [21,22,23,24,25,26,27]. Li et al. [28]. successfully prepared a ZrO_2_/WS_2_ composite coating on the surface of AISI 316L stainless steel using sol–gel technology and then prepared a biomimetic shark skin structure on the coating surface through laser texturing. The study found that when applying the biomimetic shark skin structure on the ZrO_2_/WS_2_ composite coating, lower friction coefficients, and adhesion/abrasive wear were exhibited. This phenomenon is attributed to the fact that not only can uniformly distributed diamond units reduce the contact area, but microgrooves embedded in the diamond units can also collect wear debris. In this paper, biomimetic pangolin texturing was prepared on the surface of cast steel 20Mn using a fiber laser, a high-hardness TiN coating was processed using the PVD method, and the friction and wear performance were tested. This study attempts to provide a potential method for improving the dry friction performance and hydrophobicity of the surface of cast steel 20Mn by utilizing the synergistic effect of biomimetic pangolin structural texturing and TiN coating.

## 2. Materials and Methods

### 2.1. Materials

In this study, cast steel 20Mn was selected as the base material, which was used as the material for the front-end ear-picking roller of the harvester (4YZPSJ-4C(G4), Shandong Jindafeng Machinery Co., Ltd., Dezhou, China). The samples were cut into 10 mm × 10 mm × 6 mm (length × width × height) cubes by wire cutting. The matrix chemical composition was measured by a direct reading spectrometer (LABM12, SPECTOR, North Rhine-Westphalia, Germany) as shown in Table 1. The tested samples were polished to a mirror finish. The surface roughness of the samples was measured with a roughness tester, with a value of 0.02 μm.

### 2.2. Surface Texturing Pretreatment

As depicted in Figure 1a, the body of a pangolin is enveloped by hard scales. Pangolins typically inhabit burrows in grassy foothill regions or damp areas with shrubs on hillsides. The texture present on the surface of pangolin scales potentially mitigates wear. Inspired by these pangolin scales, this research project devised a biomimetic texture that emulates the appearance of pangolin scales, as demonstrated in Figure 1b. The geometric parameters of the structure were set with a radius R = 200 μm and texture width of 50 μm. A fiber laser marking machine (Zhongtian laser, Kunshan, China) was used to create a microtexture array, as shown in Figure 2a. The laser-processing parameters were as follows: scanning speed of 100 mm/s, average power of 15 W, pulse repetition frequency of 20 KHz, 5 scanning passes, and all laser ablation processes were conducted at 25 °C and 37% humidity. Then, the surface of textured samples was polished and cleaned in an ultrasonic bath to remove the oxide layer and debris. For clarity, the polished surface without texture was known as PS, while the surface with the processed pangolin scale biomimetic texture was known as TS.

### 2.3. PVD TiN Coating Processing

As shown in Figure 2b, the TiN coating was processed on the surface of the sample using the Physical Vapor Deposition (PVD) method. PVD was used (Huafu Metal Products Co., Ltd., Dongguan City, China) to deposit TiN coatings on the surfaces of PS and TS, respectively. Pure titanium targets (purity of 99.9%) were utilized for sputtering, with 99.99% pure nitrogen injected into the sputtering chamber as the sputtering gas. The deposition pressure was 2.0 Pa, the deposition temperature was 400 °C, and the deposition time was 7 h. For clarity, the material surface processed with TiN coating without biomimetic texture was labeled as TPS, while the material surface processed with TiN coating after the biomimetic texture was applied was labeled as TTS. The thickness of the deposited coating was 3 μm.

### 2.4. Experimental Design

The sliding wear test was carried out in the Multifunctional Friction and Wear Testing Machine (MFT-5000, RTEC, San Jose, CA, USA) with the aim of investigating the impacts of biomimetic texture and Ti coating on the sliding wear behavior of the substrate. Figure 3 presents a schematic diagram of the sliding wear test. Given that the substrate material in this experiment was intended for use in the front-end ear picker roller of the corn harvester, which is a component that directly interacts with corn ears and demands a high level of wear resistance, a relatively harder GCr15 steel ball with a hardness of 55 ± 3 HRC was selected as the friction counterpart. In the actual operational context of the corn harvester ear roller, lubrication is absent. Therefore, dry friction was opted for as the experimental condition. All tests were performed without the use of lubricants at room temperature (25 °C) and 30% humidity. The experimental parameters are enumerated in Table 2. To minimize experimental errors, each test was replicated three times during the sliding wear tests.

### 2.5. Sample Characterization

The three-dimensional profile of the sample surface after lasering was characterized by the white light module of the multifunctional friction and wear tester (MFT-5000, RTEC Co., Ltd., San Jose, CA, USA). The secondary electron mode (SE) of the Field emission scanning electron microscope (Geoini300, Carl Zeiss AG, Oberkochen, Germany) was used to observe the microstructure of the sample surface, and the elemental composition of the surface was analyzed by an X-ray energy chromatograph (EDS) assembled in the SEM. The phase analysis of the modified surface was carried out under the conditions of target voltage 40 KV, current 40 mA, scanning speed 10°/min and scanning range 30–90°. A surface roughness detector (SJ-410, Mitutoyo, Kanagawa, Japan) was used to measure the surface roughness Ra, and a Vickers microhardness tester (402 MVD, Wilson Ltd., Norwood, MA, USA) was used to detect the change in surface hardness of the modified samples. The contact angle was measured using a contact angle measurement system (SDC-350, SINDIN, Dongguan, China), 3 μL of deionized water was dropped on the surface of the sample, and CA measurements were performed after the droplets stabilized for 3 s to characterize their wettability. Averaged from 5 measurements to minimize error. Wear marks were detected using an ultra-deep well (USP-Sigma, RTEC Co., Ltd., City of Saint Louis, MO, USA).

## 3. Results

### 3.1. Characteristics of the Textured and TiN-Coated Surfaces

As illustrated in Figure 4d, the three-dimensional surface morphology of the pangolin biomimetic texture following laser processing was examined. It was discerned from the image that the depth of the biomimetic texture is approximately 25 to 45 μm, with a width spanning from 45 to 55 μm, a fan-shaped radius of 200 μm, and a textured area ratio (texture area/total area) of around 30%.

Figure 4a–c display the SEM images of the modified surfaces. The image presented in Figure 4a represents the surface morphology of the biomimetic texture as observed at a magnification of 50 times. It can be observed from the image that the laser processing led to a distinct outline and a surface with a structurally regular pangolin scale-like appearance.

At the bottom of the laser irradiation trace, a layer composed of fine grains forming a remelted layer and a water-like structure is observed, while many micro-nano voids and cracks were distributed at the groove edges (Figure 4c,d), which may be related to the combined effects of laser ablation and melting [29]. During the lasering processing, surface vaporization, separation, and remelting usually occur as basic stages. The energy distribution of pulsed laser exhibits Gaussian characteristics, with a peak intensity at the center of irradiation that attenuates towards the periphery.

At the center of the laser irradiation spot, high-energy laser beams impacted the surface of the test specimen, causing material ablation, evaporation, or even explosive phenomena. Due to the high scanning laser sintering temperature and fast cooling rate, a dense remelted layer was obtained [30]. The Marangoni effect predominated in the central region of the laser texture, where the interaction between this effect and the recoil pressure caused by liquid evaporation in the melt pool led to outward liquid flow. After the laser pulse ended and the temperature decreased, the liquid solidified, resulting in protrusions on the coating surface [31].

Figure 5 depicts the SEM surface morphologies of TiN-coated samples possessing diverse surface topographies. It is evident from the figure that the coating morphology on the substrate exhibits its porous surface structure. Apparently, there exist certain defects on the surfaces of all coatings, such as cavities and surfaces with large particles. These are typical characteristics commonly found in coatings deposited by PVD on substrates that have been treated with gas abrasives [32,33]. There are pores of noticeably different sizes within the TiN coating. One potential cause for the non-uniform coating could be an unclean surface. The “large particles” formed during the deposition process are a result of the direct deposition of un-ionized film-forming particles onto the coating surface during the deposition procedure.

It is worthy of note that the two varieties of micro-textures formed on the substrate are not entirely concealed by the TiN coating, as shown in Figure 5c,d. The coating deposition procedure exerts no substantial influence on the morphology of the texture pattern. In Figure 5c, it was found that the coating quality of the laser protrusion was lower than that of the non-laser part, which was due to the increase in the surface roughness and porosity of the material after the laser.

Figure 6 shows the XRD phase analysis map. The Fe phase peaks of the matrix appear at 44.352°, 64.526°, and 81.654° on the (110), (200), and (211) crystal planes, which fully conform to the standard card number 85-1410. After laser processing of microtexture, the intensity of diffraction peaks decreases. This phenomenon is caused by the increase in surface roughness and surface grain refinement after laser processing. In XRD testing, rough surfaces can complicate the scattering of X-rays. When X-rays hit an uneven surface, in addition to normal diffraction, more diffuse scattering occurs. These diffuse X-rays can interfere with the reception of normal diffraction signals, resulting in increased background noise and larger fluctuations in the diffraction pattern; the rapid heating and cooling process of laser processing can refine the grains on the test specimen. Generally speaking, the intensity of diffraction peaks will be reduced after grain refinement. As can be seen in the figure, the characteristic diffraction peaks of the TiN phase appear on the surface of the material after processing TiN coating, and the half-height width of the diffraction peak is very narrow, indicating that the crystal growth is good. The internal defects are few, and the coating quality is high. TiN has a face-centered cubic structure, and its main diffraction peaks usually appear near the 2-theta angles of about 36.7°, 42.6°, and 61.8°, which correspond to the (111), (200), and (220) crystal planes of TiN, respectively. It can be seen in the figure that there is a preferential orientation during the growth of TiN crystals, and the (111) crystal plane is preferentially grown in the XRD pattern, and other crystal planes are covered, which is a common phenomenon in PVD processing.

Figure 7 showcases the EDS results of the material surfaces. From Figure 7a, it is clearly observable that, in addition to Fe, Si, and C, O elements are detected on the surface of the test specimen. The origin of O is mainly ascribed to oxidation reactions when the material is exposed to air. As depicted in Figure 7b, following laser processing, there was a growth in the O content on the material surface, with the weight percentage (Wt) ascending from 0.43% to 12.54%, and O is distributed at the edges of the texture. By integrating Figure 7c,d, it can be noticed that not only is the O content substantial on the TS surface but it is also enhanced on the TTS surface, signifying that oxidation took place during the femtosecond laser processing. During the processing, the laser did not inflict obvious thermal damage to the interior of the material; thus, the oxidation traces were mainly dispersed within the texture. The existence of oxides frequently results in an augmentation of material hardness, and the material’s wear resistance is also marginally improved. Cheng et al. [34] suggested that oxides may have beneficial or detrimental effects on the tribological behavior of materials. If the bond strength between oxides and the steel substrate is low, severe wear may occur due to fracturing or detachment from the steel substrate. In essence, under the action of frictional forces, adhesion points are sheared and slid, leading to oxide debris acting as free third bodies, causing wear on opposing surfaces. Ultimately, the wear process becomes a cycle of alternating adhesion and abrasion debris formation.

The coating samples were processed through wire cutting to obtain cross-sectional samples. Subsequently, they were polished with sandpaper until a mirror-like finish was attained. After that, ultrasonic cleaning was carried out using anhydrous ethanol, and then the samples were dried. The cross-sectional morphology was observed under a SEM for microscopic analysis. As illustrated in Figure 8, the cross-sectional morphology of all TiN coating samples exhibited dense microstructures without any voids. The TiN coating was effectively deposited at the bottom and edges of the microgrooves. There were no significant disparities in coating thickness among different samples, which implies a favorable adhesion interface between the coating and the substrate. Nevertheless, the interface bonding between the coating and the textured substrate is more compact. This might be attributed to the enhanced surface activity of the steel substrate resulting from surface deformation.

As shown in Figure 9b, the roughness of the PS surface Ra = 0.02 μm, the TS surface Ra = 10.30 μm, and the roughness of the TPS surface Ra = 0.21 μm also increased, and the TTS surface Ra = 8.74 μm. The increase in roughness is mainly due to the presence of microstructure and the introduction of rough coating. The TS-treated surface exhibits the highest roughness value, which is caused by the significant height difference between the bottom of the surface and the raised part. These findings indicate that the modification process not only changes the microstructure of the metal surface but also has an important impact on its surface roughness.

Hydrophobicity is one of the important properties of solid surfaces. Metal materials with high hydrophobicity have more advantages in aspects such as corrosion prevention and self-cleaning. At present, a large number of researchers have tried to apply hydrophobic surfaces to the field of friction reduction and wear resistance to study the wear resistance of hydrophobic surfaces. Yilbas et al. [11] studied hydrophobic surfaces fabricated using lasers. The results showed that the coefficient of friction was reduced compared to untreated surfaces. Hao et al. [35] further confirmed the above conclusion. Typically, the contact angle (CA) is used to characterize the wettability of solid surfaces [36]. The classification of solid surface wettability is based on different CAs, where solid surfaces are considered hydrophilic (CA < 90°), hydrophobic (90° < CA < 150°), or superhydrophobic (CA > 150°). In this study, we employed a contact angle measurement system to characterize the wettability of samples by precisely measuring the contact angle. The experimental results revealed the significant influence of surface morphology on wettability (Figure 10a). Specifically, PS surface CA = 88.89°, TS surface CA = 151.26°. Incredibly, laser microtexture makes the substrate surface change from a hydrophilic surface to a superhydrophobic surface. According to the Cassie–Baxter model [37]:(1)$$\cos\theta^{*}=f_{1}\cos\theta -f_{2}$$
where θ is the contact angle on a smooth surface, $f_{1}$ is the proportion of the actual liquid-solid contact area to the total apparent contact area, $f_{2}$ is the proportion of the liquid-air contact area to the total apparent contact area, and $f_{1}+f_{2}=1$. For a smooth surface, assuming $f_{1}=1$, $f_{2}=0$ (no air involved), it adheres to the traditional Young’s equation [38]:(2)$$\cos\theta =\left(\gamma_{sv}-\gamma_{sl}\right)/\gamma_{lv}$$
(where $\gamma_{\delta v}$ is the solid-vapor surface energy, $\gamma_{sl}$ is the solid-liquid surface energy, and $\gamma_{lv}$ is the liquid-vapor surface energy), complete wetting between the liquid and the solid occurs (Figure 10a). When the contact angle changes to $151.26^{\circ }$ after processing the texture, according to the Cassie–Baxter model, $f_{1}$ decreases and $f_{2}$ increases. This is because the texture prevents the liquid from fully contacting the solid surface, with some portion occupied by air (Figure 10c). For instance, assuming after processing the texture, $f_{1}=0.3$, $f_{2}=0.7$, the apparent contact angle $\theta^{*}$ can be calculated through Equation (Equation 1). This model effectively explains the influence of texture on the contact angle by increasing the contact angle through altering the proportion of contact between the liquid, solid, and air, transitioning the material surface from hydrophilic to hydrophobic. TPS surface CA = 79.47°, which is smaller than the PS. This phenomenon can be explained from two perspectives. On the one hand, the increase in surface roughness may lead to a change in the contact angle [39,40,41]. On the other hand, the lower the surface energy, the more hydrophobic the material surface is, leading to a larger contact angle; conversely, the higher the surface energy, the more hydrophilic the material surface is, resulting in a smaller contact angle. Due to the high surface free energy of TiN surfaces, PVD processing of TiN coatings increased the surface energy of the material surface, resulting in lower measured contact angles for all ionic liquids on TiN coatings [42,43].

The TTS surface exhibited a contact angle (CA) of 112.82°. The experimental findings demonstrated that, in conjunction with the augmentation of the surface roughness of the TiN coating, the contact angle manifested a complex pattern of variation. Within a low roughness range, the contact angle may decrease with increasing roughness, aligning with predictions from the Wenzel model. As the roughness further increases, there may be a turning point in the contact angle, after which it increases with the roughness, in accordance with the Cassie–Baxter model.

Figure 9b shows the hardness change in the modified surface. During the hardness measurement, the loading force was set to 500 N, and the holding time was 15 s. The specific measurement method was to select four uniformly distributed points on the PS surface and the edge of the bionic texture, respectively, and then take the average value. The results showed that the hardness value for the PS surface was 150.26 HV, while the hardness value for the TS was 201.17 HV, representing a 33.88% increase compared to the PS. The hardness of the TPS reached 244.10 HV, showing a 62.45% increase compared to the PS. Furthermore, the hardness value for the TTS was 318.31 HV, indicating a substantial 111.84% increase in hardness compared to the PS. After processing biomimetic micro-textures, the hardness of the substrate had significantly increased, and the improvement in surface microhardness could be attributed to grain refinement [44]. During multiple erosion processes, inevitable oxidation phenomena, as shown in Figure 9b, could occur, which could enhance the surface hardness of the material [45].

The hardness values of the TS and the TTS were significantly higher than those of the PS and TPS. This is due to the high hardness of TiN coatings, which, when applied to the material surface, can significantly enhance overall hardness performance [46,47]. TiN typically exhibits good strength and hardness characteristics. During the coating process, there may be some interaction between TiN and the base material, such as the formation of chemical bonding or the generation of diffusion layers, therefore enhancing the surface hardness of the material. For example, research by Ke Chen et al. [48] involved processing TiN coatings on stainless steel. The study indicated that high-energy processing chambers promoted the rapid formation of chemical bonds in the TiN layer, revealing the corrosion-resistant mechanism of TiN at the atomic scale. This demonstrates a strong chemical bonding between TiN and the substrate, forming a dense protective layer.

### 3.2. Friction and Wear Properties

To explore the impact of different surface pretreatment processes on the tribological performance of the substrate surface, dry friction and wear tests were conducted. Since the results of the above three experimental parameters have little difference in friction coefficient, the first set of experimental results is selected for analysis. Figure 11 clearly demonstrates the comparative results of the friction coefficients of different textured samples under dry friction conditions, providing a visual basis for subsequent analysis. As shown in Figure 11, the friction coefficients of each sample sharply increased in a short period (running-in period), and then, as the sliding time continued to increase, the friction coefficient reached a certain value and stabilized (table friction stage). This change in behavior can be explained by the Hertz contact stress theory. During the process of sliding wear, the contact area between the two contact surfaces gradually increased.

For the PS sample, as shown in Figure 11a, the friction coefficient exhibited a sharp increase in the initial stage. When the number of sliding cycles reached 132, the friction coefficient tended to stabilize, entering a stable operational phase, which could be considered a relatively short running-in period. Subsequently, the PS sample transitioned to a stable state, with an average friction coefficient of around 0.84. At the beginning of the curve, the friction coefficient was lower due to its high surface glossiness. Due to the effect of abrasion, the surface of the sample was rapidly damaged during the sliding wear process. The generated wear debris adhered to the friction surface, increasing the adhesion between sliding interfaces and resulting in an increase in the friction coefficient. Therefore, the transient increase in the friction coefficient of the PS sample could be attributed to the interaction between debris [49]. Compared to the PS sample, the TS sample had a longer running-in period, with the number of sliding cycles reaching 298, which can be considered a longer running-in phase. This was related to the reduced contact area caused by the protruding structures on the sample surface. In the stable friction stage, the TS had a friction coefficient of 0.67, lower than the stable friction coefficient of the PS (approximately 0.84). Compared to the polished substrate, the friction coefficient of TS had decreased by 20%. This was mainly due to the unique microstructure of pangolin scales, where the texture of pangolin scales typically had a certain direction and periodicity. From a materials mechanics perspective, this microstructure can alter the stress distribution on the material surface. When biomimetic textures are applied to metal or other material surfaces, it is like creating a microscopic “buffer zone” on the material surface. During friction, this texture can disperse stress, preventing stress concentration at a specific point, thus reducing instances of localized excessive wear. In the friction process, traditional smooth surfaces have large direct contact areas with the mating surface. Surfaces with biomimetic textures, however, reduce the actual contact area with the mating surface due to the presence of textures such as pits and ridges. According to the classical friction theory F=μN (where *F* is friction force, μ is the friction coefficient, and *N* is the normal force), a decrease in the actual contact area leads to a decrease in friction force under constant normal pressure. For example, consider a smooth metal block rubbing against another flat surface with a large contact area. When the surface of this metal block is textured like a pangolin, creating many small “isolated zones” on the surface where only the tops of these zones contact the surface, the actual contact area decreases, leading to a natural reduction in friction. Additionally, the reduction in the friction coefficient due to the TS treatment could be attributed to the capture and storage behavior of wear debris in the microgrooves, alleviating third-body wear between the contact areas of the mating surfaces.

Finally, as analyzed earlier, the TS surface had a higher hardness than the PS surface, which was also a significant factor contributing to the reduction in the friction coefficient. Therefore, surface deformation pretreatment can effectively enhance the anti-friction performance of the substrate surface. Figure 9b depicts the variation in friction coefficients of samples after deposition of TiN. Similarly, all samples exhibited a running-in phase, with the TPS surface having a running-in cycle count of 82 cycles, followed by a stable friction coefficient phase. In the stable friction stage, the average friction coefficient of the TPS sample 0.58 decreased by 30.9% compared to the PS surface. This was mainly due to the increased surface hardness after being processed with a TiN coating, making it less prone to being scratched by steel balls on the metal surface, reducing the contact area and, therefore, lowering the friction coefficient. The TTS surface had a running-in cycle count of 177 cycles. In the stable stage, the average friction coefficient of the TTS sample 0.52 in the stable sliding phase decreased by 38.1% compared to the PS sample. This result was a collaborative outcome of the TiN coating and pangolin biomimetic texture, indicating that surface texture pretreatment played a positive role in enhancing the tribological performance of TiN coatings.

To better study the wear mechanism, the wear surface of the material after the first group of tests was analyzed by SEM. The morphology of the worn surface was observed in the middle of the wear track on each sample. In the PS sample (Figure 12a), many particles, such as wear debris and loose fragments, were distributed on both sides of the wear track. Additionally, a large amount of adhesion material, wear debris, delamination, and mechanical plowing phenomena were observed in the central part of the worn surface. This indicated severe adhesive wear on both surfaces, where adhesive points were formed due to plastic deformation in the contact area between the test specimen and the hard GCr15 steel. As the reciprocating sliding continued, these adhesive points underwent shear fracture due to significant frictional heat, leading to the formation of wear debris. Research indicated that in adhesive wear if the wear debris was hard and free, abrasive wear could occur [50]. (Figure 12a) Abrasive wear debris and particles were prominently distributed along the wear track, indicating the presence of significant abrasive wear in the contact area. As the GCr15 steel ball rubbed against the friction surface, it began to generate fragments and particles. With continued friction, due to increased frictional force and heat, adhesion occurred again. When a large number of abrasives come into play, material loss increased. Under the repeated action of abrasive particles, layer-by-layer wear mechanisms led to the formation of wear debris. This adhesion and hard abrasive wear debris increased the likelihood of stress concentration, reducing the surface strength. This wear behavior confirmed that the primary wear mechanisms of the steel matrix were abrasive wear and adhesive wear, accompanied by a small amount of fatigue wear.

In Figure 13(a1), EDS elemental analysis showed a large number of oxygen elements detected in the middle of the wear track, confirming oxidation wear during dry air friction tests. In fact, under the action of continuous reciprocating sliding, the contact area was in a plastic flow state, and the generated oxides became the bonding material on the contact interface. Similarly, in this scenario, the generated oxides would act as bonding material aside from the lubricating material. Figure 11a, the PS sample with the most severe oxidation adhesion had the highest friction coefficient. Figure 13(a2) further confirmed the adhesion of iron elements in the wear track by the ball.

In comparison, the wear traces in the middle of the TS worn surface were significantly reduced (Figure 12b). Within the wear area, wear mainly occurred on the hardened layer at the edges of the laser texture, with only slight scratches and some laser protrusion peeling off onC the matrix surface. This indicated that biomimetic texturing reduced the abrasive wear of the material. At the beginning of reciprocating dry sliding, the strengthening layer (laser protrusions) predominantly affected the wear resistance of frictional wear. The strengthening layer underwent slight deformation during friction, and due to its high hardness, there was less wear debris and fewer grooves. With increased reciprocating sliding, the strengthening layer and micro-textures were worn, generating abrasive particles that aggregated to form a third body, leading to three-body wear. At this point, the microtexture pits captured wear debris, reducing the actual contact area and shear stress, therefore decreasing abrasive wear. Therefore, no significant wear debris was found on the worn surface of the TS sample, while a large amount of wear debris was distributed within the texture. Additionally, the presence of a hardened layer with higher microhardness reduced adhesion.

In Figure 13(b1), EDS elemental analysis showed that the O elements generated during laser texturing decreased after a period of wear. The O elements on the unaltered matrix surface showed minimal variation, proving that the wear primarily occurred at the textured protrusions during the wear process. The distribution of Fe elements in Figure 13(b1) confirmed the reduction in adhesive material.

Figure 12c exhibited the SEM wear morphology of the TPS sample subsequent to 1500 sliding cycles after the application of a TiN coating. It was clearly discernible from the image that the sample surface remained intact, devoid of any indications of plowing, spalling, or the presence of wear debris within the wear tracks. However, a substantial amount of adherent material was observable on the surface. It is widely acknowledged that a prominent characteristic of GCr15 bearing steel is its intrinsic viscosity during sliding friction, which is manifested as a propensity for material transfer. The high-hardness TiN coating further expedited the wear of the steel ball with which it interacted. The EDS image of the abraded surface, as presented in Figure 14, disclosed the distribution of Fe elements, as depicted in Figure 14b. This further corroborated the adhesion of the wear ball within the wear track, providing compelling evidence for the analysis of the wear mechanism.

Figure 12d shows the SEM wear morphology of a TTS sample with a textured surface after applying a TiN coating after 1500 sliding cycles. It is clear that the wear of the TTS surface occurs at the position of the laser protrusion, and the protrusion will undergo plastic deformation during repeated extrusion. After 1500 repeated rubbing, the laser pits are nearly filled in the more severe positions, but it is undeniable that the smooth substrate surface without the laser is intact at this time, showing a complete appearance without any scratches or wear marks, which is the role of the laser protrusion. In the early stage of wear, due to the laser bumps, the smooth substrate surface does not participate in friction, and trace amounts of abrasive particles will occur in the bumps. At this time, trace abrasive particles will be captured by the laser pits. After a period of friction, the laser bumps will produce plastic deformation, and the pits will gradually be filled. The laser bumps that drop wear debris in the early stage become part of the substrate due to repeated extrusion in the laser pits, and since the pits are filled with laser bumps and coating materials, this will make the material in the pits higher than the original hardness of the substrate. The above results show that the most wear-resistant surface is obtained through the synergy of laser processing and TiN coating. The distribution of N and Ti elements in Figure 15c,d shows that the TiN coating on the surface of the sample remains intact after wear without observable loss. However, there is some loss of TiN coating on the edges of the textured bumps, indicating that these bumps are involved in the friction process. Figure 16 is a diagram of the wear mechanism, from which the wear process can be seen. Figure 17 is a three-dimensional map of different surface wear topographies, further confirming the previous analysis.

## 4. Conclusions

In this study, TiN coating and bionic pangolin scale texture were successfully applied to the 20Mn metal surface of agricultural cast steel to improve its wear resistance and hydrophobicity, and a series of significant results were achieved.

In terms of wear resistance, the pin-disc wear test showed that the surface of the cast steel 20Mn modified by TiN coating and bionic scale texture exhibited excellent wear resistance. TiN coating provided high-hardness basic protection and effectively resisted the cutting action of abrasive particles, while the bionic texture reduced abrasive wear and fatigue wear during the wear process by optimizing the tribological characteristics of the surface, reducing the friction coefficient and stress concentration phenomenon. Compared with the untreated test specimen, the friction coefficient of the composite modified sample was significantly reduced, which was about 62% of the original sample.

In terms of hydrophobicity, the PS surface CA = 88.89°, TS surface CA = 151.26°, and the transition from hydrophilic surface to superhydrophobic surface was completed after processing the bionic microtexture of pangolin. However, due to the high surface energy of TiN coating, the TPS surface CA = 79.47° and the TTS surface CA = 112.82°.

## Figures and Tables

**Figure 1 biomimetics-10-00042-f001:**
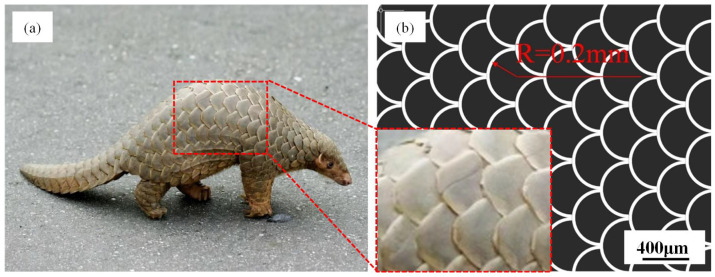
(**a**) Pangolin Scale Structure, (**b**) Biomimetic Texture Design.

**Figure 2 biomimetics-10-00042-f002:**
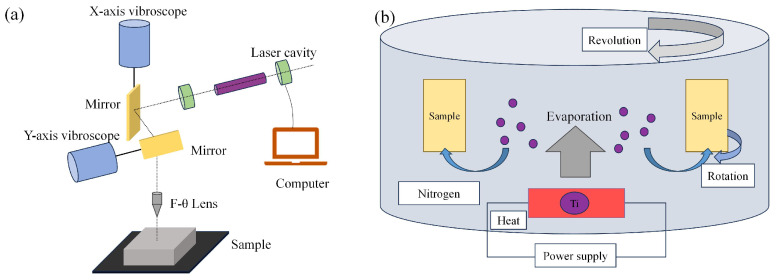
(**a**) Fiber laser. (**b**) TiN Coating Processing Schematic.

**Figure 3 biomimetics-10-00042-f003:**
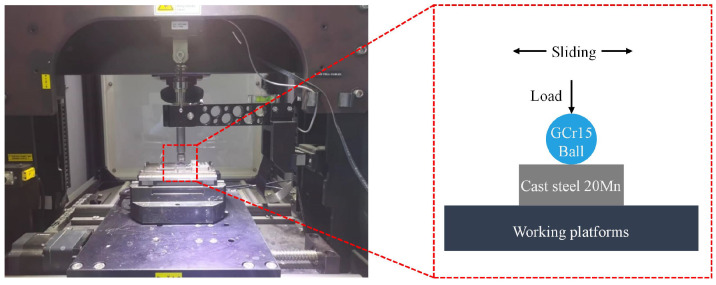
Illustration of surface wear.

**Figure 4 biomimetics-10-00042-f004:**
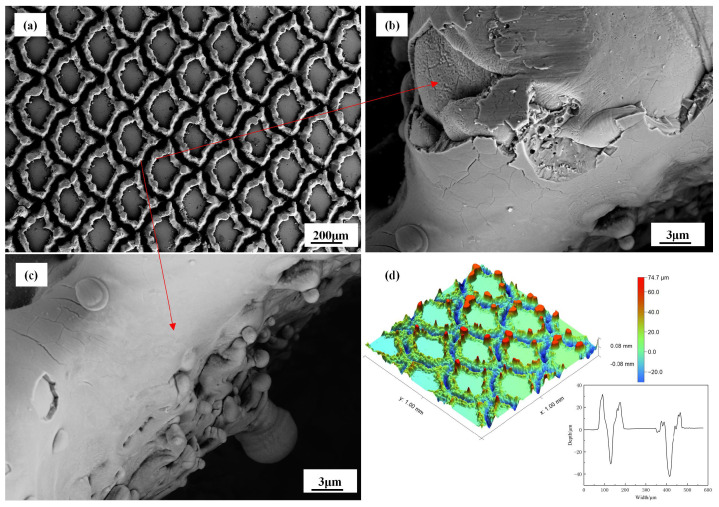
(**a**–**c**) are SEM images of laser-textured structures. (**d**) is a super depth of field three-dimensional morphology image of laser-textured structures.

**Figure 5 biomimetics-10-00042-f005:**
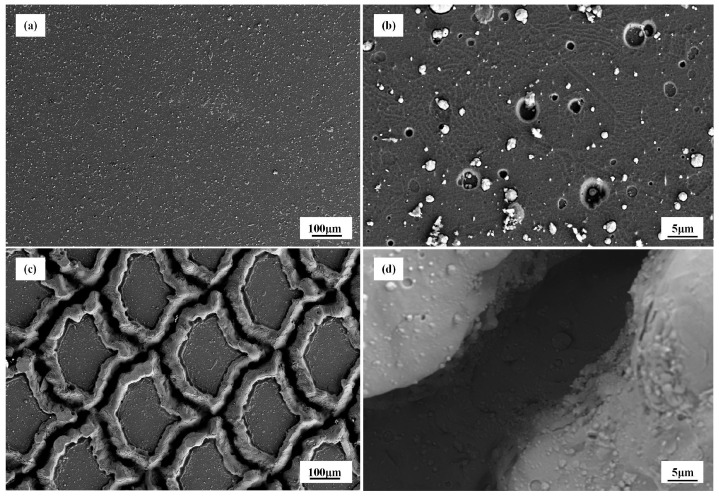
(**a**,**b**) TPS surface SEM morphology. (**c**,**d**) TTS surface SEM morphology.

**Figure 6 biomimetics-10-00042-f006:**
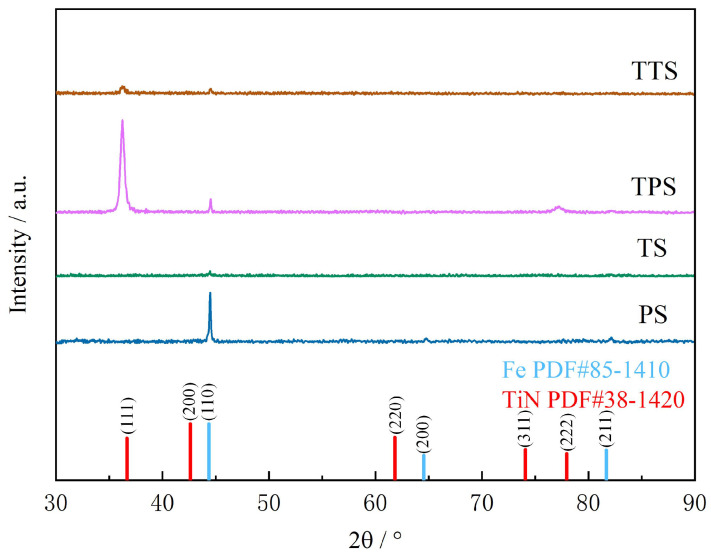
X-ray diffraction patterns of test samples.

**Figure 7 biomimetics-10-00042-f007:**
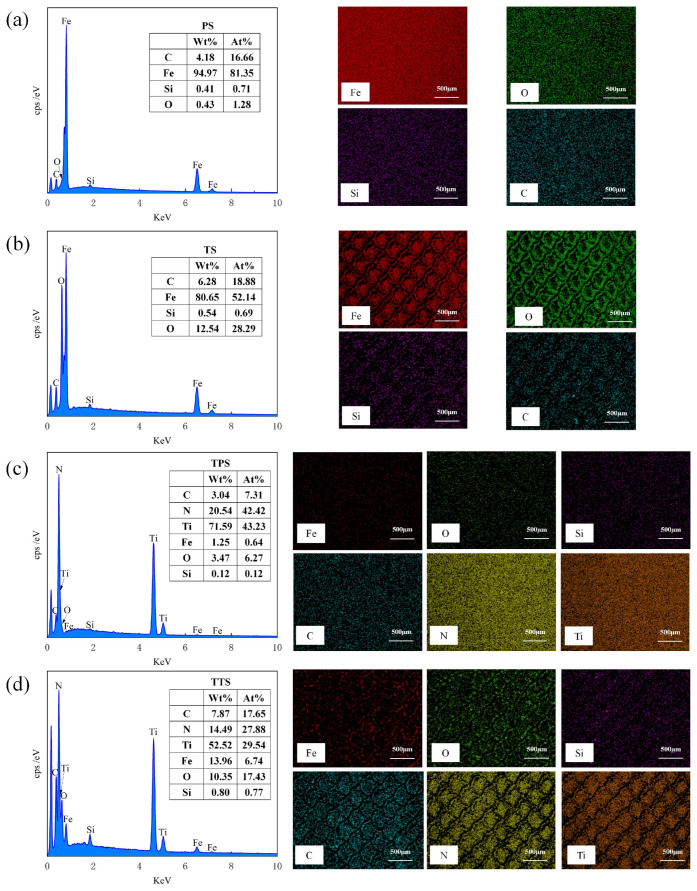
Elemental composition analysis using EDS (**a**) PS, (**b**) TS, (**c**) TPS, (**d**) TTS.

**Figure 8 biomimetics-10-00042-f008:**
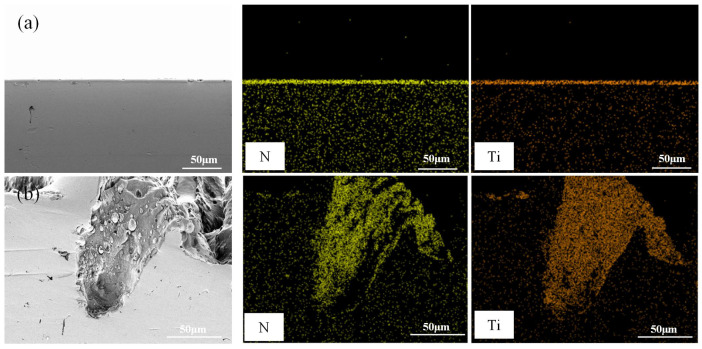
SEM micrographs and EDS mapping of the sample cross-sections (**a**) TPS, (**b**) TTS.

**Figure 9 biomimetics-10-00042-f009:**
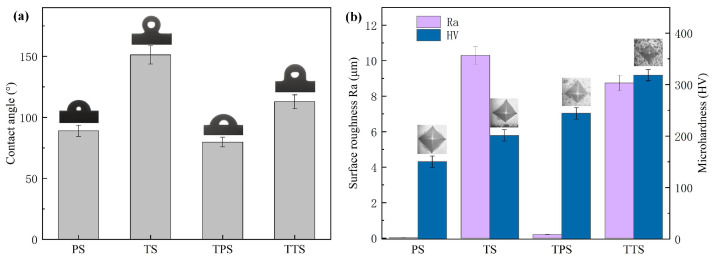
After surface modification, (**a**) average contact angle measurement, (**b**) average surface hardness, and average surface roughness.

**Figure 10 biomimetics-10-00042-f010:**
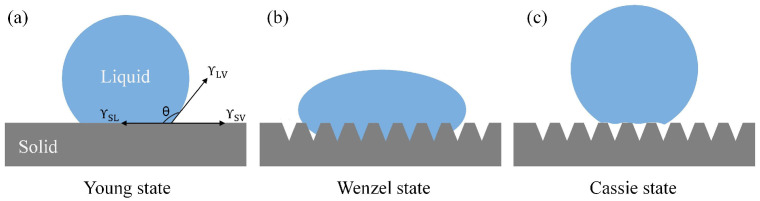
Wetting behavior of droplets on various solid surfaces. (**a**) Young’s model, (**b**) Wenzel model, (**c**) Cassie.

**Figure 11 biomimetics-10-00042-f011:**
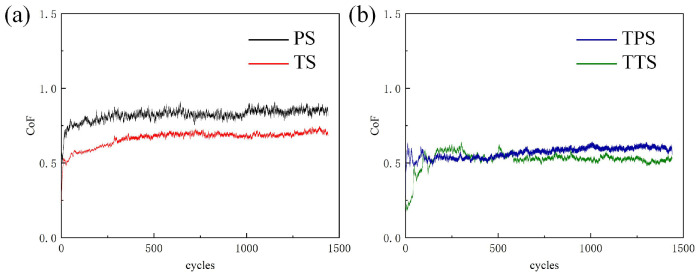
Friction coefficient variations of different textured samples (**a**) without TiN coating, (**b**) with TiN coating.

**Figure 12 biomimetics-10-00042-f012:**
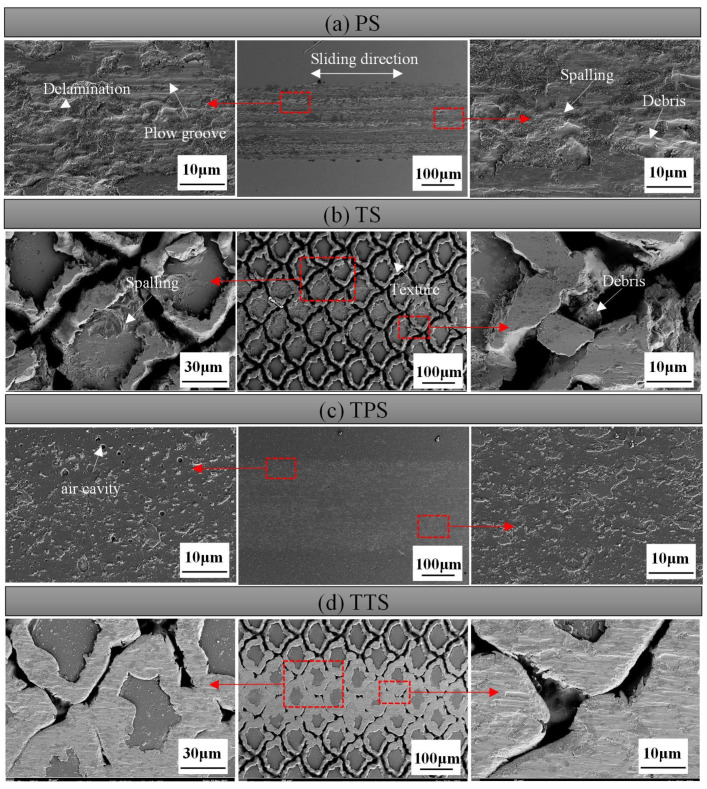
SEM Micrographs of Material Surfaces after 1500 Sliding Cycles: (**a**) PS, (**b**) TS, (**c**) TPS, (**d**) TTS.

**Figure 13 biomimetics-10-00042-f013:**
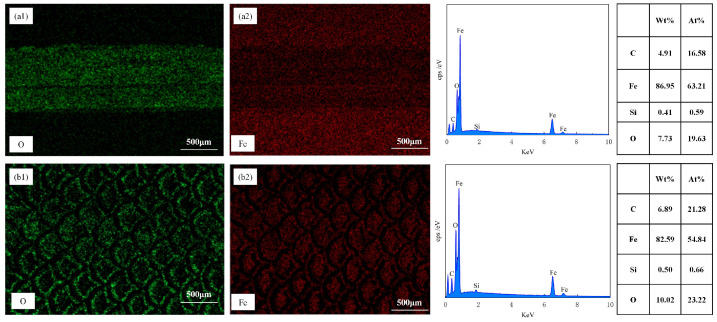
EDS Elemental Composition Analysis of Material Surfaces after 1500 Sliding Cycles: (**a1**,**a2**) PS, (**b1**,**b2**) TS.

**Figure 14 biomimetics-10-00042-f014:**
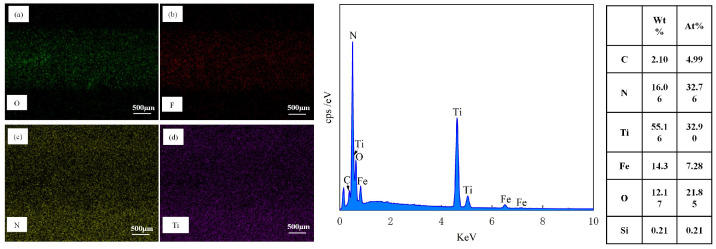
EDS Elemental Composition Analysis of TPS Material Surface after 1500 Sliding Cycles, (**a**) O, (**b**) F, (**c**) N, (**d**) Ti.

**Figure 15 biomimetics-10-00042-f015:**
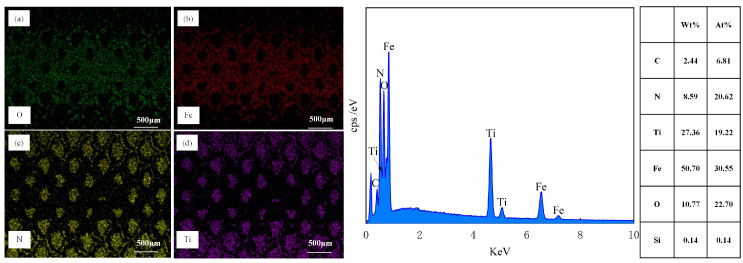
EDS Elemental Composition Analysis of TTS Material Surface after 1500 Cycles, (**a**) O, (**b**) F, (**c**) N, (**d**) Ti.

**Figure 16 biomimetics-10-00042-f016:**
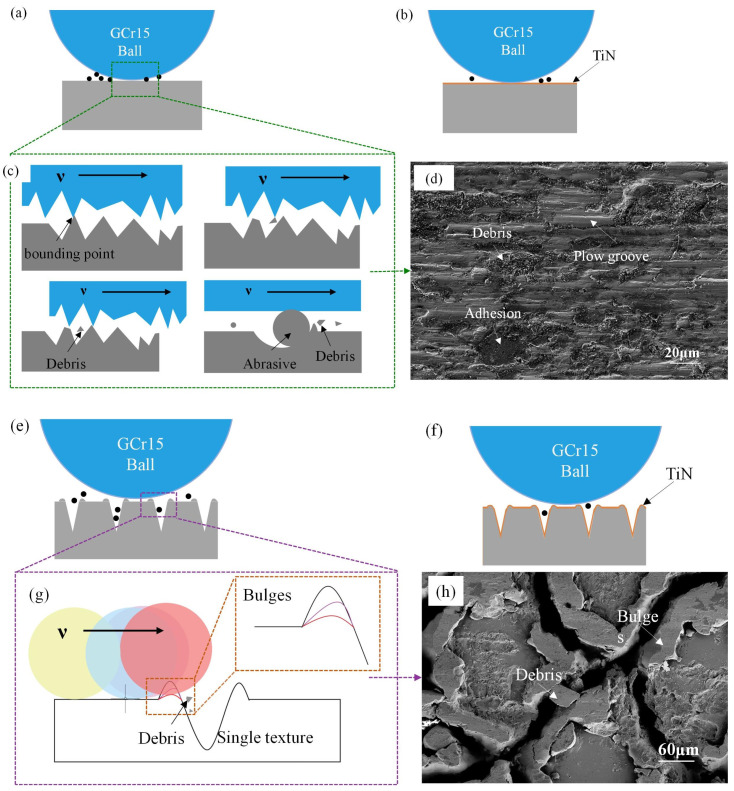
Cross-sectional schematic diagrams of the wear mechanism models of the samples under dry friction conditions: (**a**–**c**,**e**–**g**). SEM morphology images of the material surface after 1500 sliding cycles: (**d**) PS, (**h**) TS.

**Figure 17 biomimetics-10-00042-f017:**
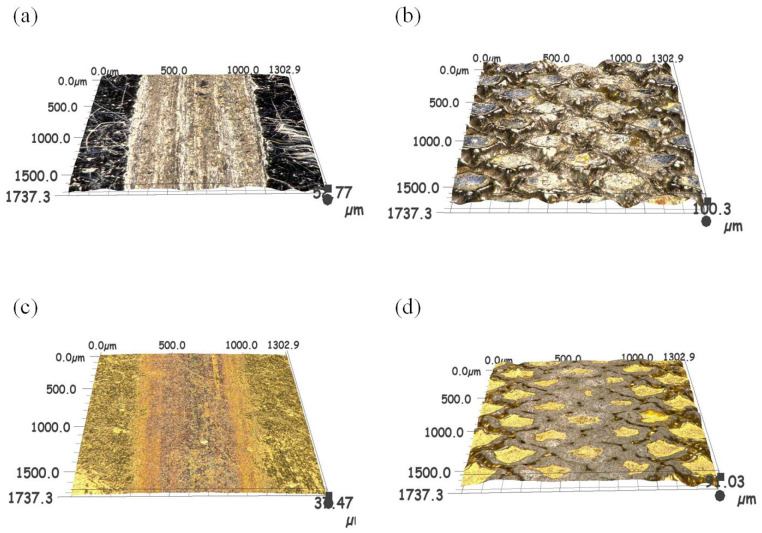
Three-dimensional morphology of wear marks: (**a**) PS, (**b**) TS, (**c**) TPS, (**d**) TTS.

**Table 1 biomimetics-10-00042-t001:** Chemical composition of 20Mn cast steel (mass%).

Elemen1	C	Mn	Fe	P	Si	S
Limits%	0.447	0.712	98.3	0.0315	0.357	0.059

**Table 2 biomimetics-10-00042-t002:** Sliding wear test parameter.

Serial Number	Working Condition	Normal Load (N)	Displacement (mm)	Cycles	Frequency of Test (Hz)
1	Dry	4	5	1500	1
2	Dry	5	5	1500	2
3	Dry	5	5	1500	1

## Data Availability

The data that support the findings of this study are available within the article and from the corresponding author upon reasonable request.

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
