# Peer review of "Utilization of TiN and the Texture of Bionic Pangolin Scales to Improve the Wear Resistance of Cast Steel 20Mn Metal"

_biomimetics, 2025, doi:10.3390/biomimetics10010042_

Round 1
Reviewer 1 Report
Comments and Suggestions for Authors
1. There is an extra equal sign on Line 37
2. The abbreviation "LSTP" on line 39 has not been referred to its full form.
3. Figure 1b should correspond to a physical enlargement of the red-framed area in Figure 1a, rather than a design drawing of the processing pattern. The design pattern appears to be similar to the scales of any kind of fish, but the connection to pangolin scales is unclear.
4. Table 2 has only one set of data. The sliding wear test should be conducted under different parameters to be convincing.
5. The relationship between hydrophobic surface and the conclusion “hydrophobic metal materials, impurities such as corn sap and shredded leaves will not easily adhere to the surface of the snapping rolls” on Line 240 has not been clearly elucidated. The practical benefits of improved hydrophobicity for corn harvesters need to be demonstrated by designing experiments or citing relevant literature.
6. The number of the subgraph in Figure 12 is confused, and the caption "Ti" in the figure should be corrected to "TiN".
Reviewer 2 Report
Comments and Suggestions for Authors
This paper is regarding the effects of TiN and the texture of bionic pangolin on the microstructural, mechanical and tribological characteristics of cast steel 20Mn. Fiber laser processing technology was employed to fabricate pangolin bionic micro-textures on the material surface, and PVD technology was utilized to deposit a TiN coating. The results showed that compared with the untreated test specimen, the friction coefficient of the composite modified sample was significantly reduced, which was about 62% of the original sample.
Following are the comments regarding this paper:
1. Pangolin Scale Structure is optimized for friction reduction in soil. Hence, it is not an ideal texture for tribological applications for cast steel 20Mn when detaching the stems from the ears of crops. Also, Pangolin Scale Structure is optimized for moving mostly in one direction, namely forward. Hence the texture is not isotropic. But for improving the adhesion of the TiN coating to the substrate, one would desire an isotropic texture. In this regard, pangolin scale structure is not the proper texture. Further, the pangolin's scales are very smooth, with each scale covering the top of another. Such overlapping structure allows the scales to form a solid defensive layer. Such structures do not improve friction performance.
2. Surface roughening is commonly used to improve the adhesion between coating and substrate. Pangolin scale structure used in this work is only one of numerous textures that could be used for this purpose. In fact, the process of fabricating the pangolin scale structure texture on the substrate is not economical compared to other more effective methods. Hence, though using pangolin scale structure texture to improve the tribological characteristics and improve the adhesion between the substrate and coating may seem to be original, it is not the best choice.
3. The TiN coatings were deposited on cast steel 20Mn substrates. What were the motivations behind choosing TiN coatings?
4. The laser micro-textured surfaces should be polished and ultrasonically cleaned to remove slags before coating TiN. The high-roughness substrate will hinder coating adhesion and will increase wear and tear. In Fig. 8b, the coating is not discontinuous, why?
5. Given the size of the textures microhardness measurements would depend strongly on the location of the tip with respect to the texture geometry. The authors should provide an image of the indentation mark to verify the validity of the hardness data given in Fig. 9b.
6. the laser textures will increase the compressive stresses and surface energies of substrates, and result in the improved adhesion strength between TiN coatings and substrates. The authors should analyze the bonding strength between TiN coating and substrate, and provide quantitative results.
7. What is the sliding distance for the dry wear test? Wear rate would be a better way to quantify wear behavior. Please provide these results.
8. The texts in Fig.4d are not clear.
Comments on the Quality of English LanguageThere are also issues with the English/grammar. And the amount of wording duplication in the manuscript should be decreased.
Reviewer 3 Report
Comments and Suggestions for Authors
In this paper, the authors investigated the influence of laser surface texturing and TiN coating on the surface hydrophobicity, friction coefficient and wear behavior. This paper has a potential to be accepted, but some important points need to be clarified or fixed.
In the following, there are the questions that the authors should answer:
1. Dramatic bulges around the texture grooves were observed. Although the bulges around textures might show higher hardness, their porosity always reduce the strength of the bulged material and the integrity of the coatings. This might restrict the effect of coatings. As an evidence, Figure 5 showed that the bulges were not fully concealed by the coating. This was further confirmed by the test results in Page 14, that the TiN coating at the contact area (bulge plateau) for TTS is worn out, rather than “intact” as described by the authors. Why do the authors keep the bulges rather than grind them away like many of the studies?
2. In this paper, the authors examined the surface hydrophobicity, friction and wear behavior. The results suggest that the TS has the highest contact angle, TPS has lowest wear, and TTS has the lowest friction. The title is named “Utilize TiN and the texture of bionic pangolin scales to improve the wear resistance of cast steel 20Mn metal”, indicating that wear performance is the most important issue for the authors. Why do the authors introduce textures to the coated surfaces?
3. The authors examined the hydrophobicity of test samples. How do the authors think about the effects of hydrophobicity to tribological behaviors in this study?
Some other detailed questions are as listed below.
Page 2 Table 1: Please check the composition of 20Mn for Carbon.
Page 3 Line 87: 20 Hz or 20 kHz?
Page 3 Line 111: 55 HV or HRC?
Page 3 Figure 2 (a): Is it a Femtosecond laser as shown in Figure?
Page 4 Table 2: Please provide the reciprocating frequency of test.
Page 5 Figure 4: It would be best to provide Interferometer or confocal images for the 3d topography as the resolution of a numerical microscope is relatively low.
Page 9 Figure 9 shows TPS has the second highest hardness while in Page 10 it states that TS is 391 HV and TPS is 354 HV. How do the authors explain this?
Page 13 Figure 12: b for the TS, TPS and TTS?
Round 2
Reviewer 1 Report
Comments and Suggestions for Authors
1. Table 2 should include all sliding wear test data, and the reason why only one set of data was selected needed to be explained in main text.
2. The inset in Figure 2b requires a scale bar. Are the scales of pangolins and the designed pattern of the same size? Is there any literature indicating that structures scaled proportionally have the same functionality?
Reviewer 3 Report
Comments and Suggestions for Authors
The manuscript has reached the minimum standard for publication.
Author Response
Dear Reviewer:
We sincerely appreciate your evaluation of our manuscript. This confirmation is an important milestone for us. We are ready to proceed with the next steps as required by the journal.
Thank you for your valuable comments.
Best regards.